# Disruption of PIKFYVE causes congenital cataract in human and zebrafish

**Shaoyi Mei[1†], Yi Wu[2†], Yan Wang[3†], Yubo Cui[4], Miao Zhang[1], Tong Zhang[1], Xiaosheng Huang[1], Sejie Yu[4], Tao Yu[2]\*, Jun Zhao[4]\***

[1]Shenzhen Eye Institute, Shenzhen Eye Hospital Affiliated to Jinan University, Shenzhen, China; [2]Shenzhen Key Laboratory for Neuronal Structural Biology, Biomedical Research Institute, Shenzhen Peking University-The Hong Kong University of Science and Technology Medical Center, Shenzhen, China; [3]State Key Laboratory of Ophthalmology, Zhongshan Ophthalmic Center, Sun Yat-sen University, Guangzhou, China; [4]Department of Ophthalmology, Shenzhen People's Hospital (The Second Clinical Medical College, Jinan University; The first Affiliated Hospital, Southern University of Science and Technology), Shenzhen, China

**\*For correspondence:**
tyu@connect.ust.hk (TY);
doctorzhaojun@163.com (JZ)

[†]These authors contributed equally to this work

**Competing interest:** The authors declare that no competing interests exist.

**Abstract** Congenital cataract, an ocular disease predominantly occurring within the first decade of life, is one of the leading causes of blindness in children. However, the molecular mechanisms underlying the pathogenesis of congenital cataract remain incompletely defined. Through whole-exome sequencing of a Chinese family with congenital cataract, we identified a potential pathological variant (p.G1943E) in *PIKFYVE*, which is located in the PIP kinase domain of the PIKFYVE protein. We demonstrated that heterozygous/homozygous disruption of PIKFYVE kinase domain, instead of overexpression of *PIKFYVE*[G1943E] in zebrafish mimicked the cataract defect in human patients, suggesting that haploinsufficiency, rather than dominant-negative inhibition of PIKFYVE activity caused the disease. Phenotypical analysis of *pikfyve* zebrafish mutants revealed that loss of Pikfyve caused aberrant vacuolation (accumulation of Rab7[+]Lc3[+] amphisomes) in lens cells, which was significantly alleviated by treatment with the V-ATPase inhibitor bafilomycin A1 (Baf-A1). Collectively, we identified *PIKFYVE* as a novel causative gene for congenital cataract and pinpointed the potential application of Baf-A1 for the treatment of congenital cataract caused by PIKFYVE deficiency.

## Editor's evaluation

This manuscript will be of interest to readers in the field of eye development and pathology and ocular geneticists. Using a zebrafish model, the authors have identified a new autosomal dominant mutation in the kinase domain of the human phosphoinositide kinase gene PIKFYVE and assessed its impact on congenital cataract (clouding of the lens). They demonstrated that heterozygous or homozygous disruption of the PIKFYVE kinase domain, but not the overexpression of PIKFYVE G1943E, led to a cataract defect, suggesting the mechanism of haploinsufficiency rather than dominant-negative inhibition of PIKFYVE activity. The authors have presented strong evidence that mutations in the PIKFYVE kinase domain could lead to congenital cataract. Overall, the same pipeline of in vitro and in vivo genetic approaches could be applied to other human genetic disorders.

## Introduction

Congenital cataract is partial or complete opacification of the lens that occurs at birth or during the first decade of life. It is a common ocular abnormality that causes visual impairment and blindness during infancy (*Khokhar et al., 2017*), accounting for 5–20% blindness in children worldwide

(*Sheeladevi et al., 2016*). According to the location and shape of the lens opacities, congenital cataract can be divided into seven clinical types: nuclear cataract, polar cataract, lamellar cataract, nuclear with cortical cataract, cortical cataract, sutural cataract, and total cataract (*Zhai et al., 2017*). Different types of cataracts cause different levels of visual impairment in patients. A variety of factors, including gene variants that affect lens metabolism. are closely associated with cataract (*Li et al., 2020*), and autosomal dominant congenital cataract is the most common mode of inheritance (*Berry et al., 2020a*). To date, the occurrence of congenital cataract has been linked with genetic variants in at least 50 genes involved in lens structure and development (*Berry et al., 2020b*; *Shiels and Hejtmancik, 2017*), including crystallin genes (*CRYAA*, *CRYAB*, *CRYBB1*, *CRYBB2*, *CRYBB3*, *CRYBA1/A3*, *CRYBA2*, *CRYBA4*, *CRYGC*, *CRYGD,* and *CRYGS*) (*Bhat, 2003*; *Zhuang et al., 2019*), membrane protein genes (*GJA3*, *GJA8*, *MIP,* and *LIM2*) (*Berry et al., 2000*; *Beyer et al., 2013*; *Pei et al., 2020*), growth and transcription factor genes (*PITX3*, *MAF,* and *HSF4*) (*Anand et al., 2018*), beaded filament structural protein genes (*BFSP1* and *BFSP2*) (*Song et al., 2009*), and other genes (*CHMP4B* and *EPHA2*) (*Dave et al., 2016*; *Shiels et al., 2007*). Although these findings have made tremendous contributions to our understanding of the genetic etiology of congenital cataract, new causative genes as well as the underlying molecular mechanisms remain to be discovered (*Berry et al., 2020a*).

In this study, we identified a missense genetic variant (p.G1943E) in the phosphatidylinositol phosphate kinase (PIPK) domain of PIKFYVE (phosphoinositide kinase, FYVE-type zinc finger containing) from a Chinese cataract family. The human *PIKFYVE* gene encodes a large protein consisting of 2098 amino acids, which contains six evolutionarily conserved domains (*Kawasaki et al., 2012*), including zinc finger phosphoinositide kinase (FYVE) domain, plextrin homology domain, β-sheet winged helix DNA/RNA-binding motif, cytosolic chaperone CCTγ apical domain-like motif, spectrin repeats (SPEC), and C-terminal fragment PIPK domain (*Li et al., 2005*; *Shisheva, 2008*). The N-terminal FYVE domain can bind to PtdIns3P on the membrane, while the C-terminal is the kinase domain which has the catalytic function for the production of PtdIns(3,5)P$_2$ (*Shisheva et al., 1999*). The in-vivo function of PIKFYVE in the development of lens and its association with congenital cataract have not been documented. Different from previously reported *PIKFYVE* variants affecting the chaperonin-like domain that cause fleck corneal dystrophy (CFD), a disease characterized by tiny white flecks scattered in corneal stroma (*Gee et al., 2015*; *Kawasaki et al., 2012*; *Li et al., 2005*), the p.G1943E variant in the kinase domain of PIKFYVE was revealed for the first time to cause congenital cataract in human patients. Using zebrafish models, we demonstrated that disruption of the PIPK domain in Pikfyve caused early-onset cataract defect (aberrant accumulation of Rab7$^+$Lc3$^+$ amphisomes in the lens) and treating the *pikfyve*-deficient embryos with V-ATPase inhibitor bafilomycin A1 (Baf-A1) significantly alleviated the vacuolation defect in the lens of zebrafish mutants.

## Results

### The congenital cataract family and clinical characteristics

We recruited a large Chinese Korean family affected with congenital cataract, which consists of 31 family members including 12 affected individuals across four generations (*Figure 1A*). All the family members were Chinese Korean except one married-in individual (III-8) who was Han Chinese. There was no consanguineous marriage in this family. Bilateral cataract was observed in every generation of the pedigree and affected parents transmitted the disease to both males and females, suggesting an autosomal dominant inheritance. No other ocular and systemic abnormalities were found in these patients. Clinical information was obtained from all patients except two deceased individuals (*Table 1*). Seven of the 10 living patients were diagnosed as cataract in their first decade of life, with the smallest age of diagnosis at 2 years old. The best corrected visual acuity (BCVA) of these patients ranged from 20/125 to 20/20. The proband (III-9) showed nuclear pulverulent cataract in both eyes (*Figure 1B*) with vision loss (OD: 20/40, OS: 20/66) since childhood. The mother (II-4) of the proband also had nuclear pulverulent cataract in both eyes, while his father (II-3) had no cataract in either eye. The son (IV-5) of the proband had Y-sutural cataract. Among the other affected members of this family, individuals II-8, III-5, III-6, and IV-3 presented nuclear pulverulent opacity in both eyes, whereas individual III-1 and his son (IV-1) showed peripheral cortical punctate opacity in both eyes. All patients of this family did not show any abnormities in fundus photography and optical coherence tomography (OCT) examinations.

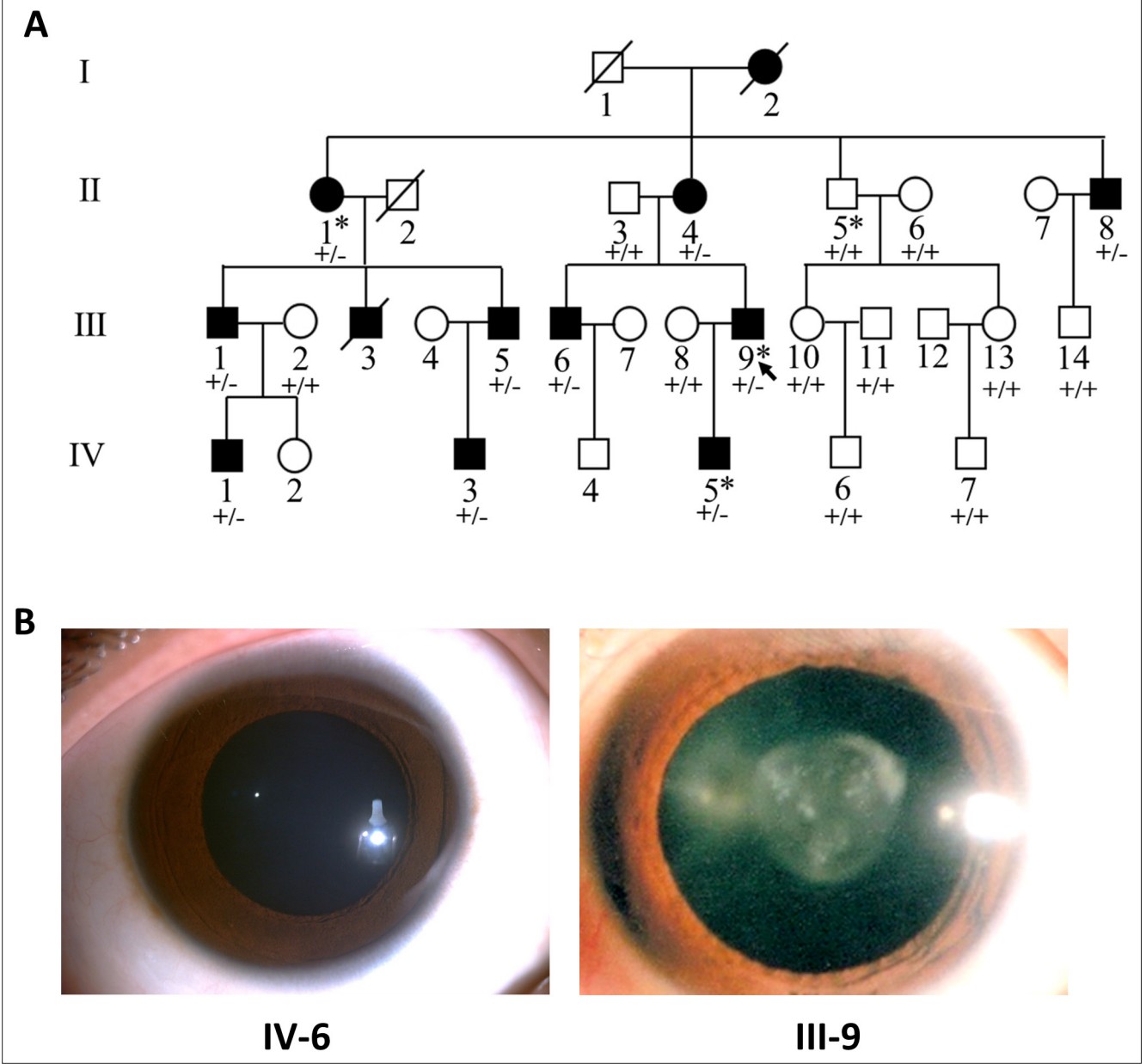

**Figure 1.** Pedigree structure and ocular manifestations of the cataract family. (**A**) Pedigree of the family with congenital cataract. Squares denote males and circles denote females; Symbols crossed by a line indicate deceased individuals. Filled symbols indicate affected individuals, while open symbols indicate unaffected individuals. All affected family members had bilateral congenital cataract. The arrow denotes the proband. The individuals marked with an asterisk (∗) are analyzed by whole-exome sequencing. Genotypes of the *PIKFYVE* variant (p.G1943E) are indicated below each symbol (+, wild-type allele; −, p.G1943E variant allele). (**B**) Slit-lamp photographs showing the transparent lens of an unaffected individual (IV-6) and the nuclear pulverulent cataract in the left eye of the proband (III-9).

## Identification of a pathogenic genetic variant in *PIKFYVE* in the cataract family

To identify the pathogenic variant responsible for congenital cataract in this family, we performed whole-exome sequencing (WES) analysis of four selected family members including three affected members (II-1, III-9, and IV-5) and one unaffected member (II-5; *Figure 1A*). Overall, 101,852–102,401 sequence variants were identified in each individual. Nonsynonymous or splicing-site variants, of which the minor allele frequencies were less than 1% in the genome aggregate database (gnomAD; https://gnomad.broadinstitute.org/) and were shared among all three affected members but were

**Table 1.** Clinical characteristics of living patients in the cataract family.

| Patient | Gender | Age at diagnosis (years) | BCVA before surgery (OD, OS) | Cataract type (OU) | Surgery (Y/N) | BCVA after surgery (OD, OS) |
|---|---|---|---|---|---|---|
| II-1 | F | 9 | – | – | Y | 20/160, 20/125 |
| II-4 | F | 12 | 20/40, 20/32 | Nuclear pulverulent | N | NA |
| II-8 | M | 4 | 20/100, LP | Nuclear pulverulent | Y | 20/40, FC |
| III-1 | M | 47 | 20/25, 20/20 | Peripheral cortical punctate | N | NA |
| III-5 | M | 6 | 20/40, 20/40 | Nuclear pulverulent | Y | 20/20, 20/20 |
| III-6 | M | 7 | 20/40, 20/50 | Nuclear pulverulent | Y | 20/16, 20/16 |
| III-9 | M | 6 | 20/40, 20/66 | Nuclear pulverulent | Y | 20/28, 20/25 |
| IV-1 | M | 22 | 20/20, 20/20 | Peripheral cortical punctate | N | NA |
| IV-3 | M | 3 | 20/30, 20/25 | Nuclear pulverulent | N | NA |
| IV-5 | M | 2 | 20/125, 20/125 | Nuclear and Y-sutural | N | NA |

BCVA: best corrected visual acuity. OD: eye, oculus dexter. OS: eye, oculus sinister. OU: eye, oculus uterque. Y: yes. N: no. F: female. M: male. LP: light perception. FC: finger counting. —: unknown. NA: not applicable.

**Table 2.** Candidate variants identified from whole-exome sequencing in the cataract family.

| Gene | SNP ID | Chromosome position (bp; hg19) | cDNA change | Amino acid change | MAF (gnomAD) | PolyPhen | GERP | CADD |
|------|--------|--------------------------------|-------------|-------------------|--------------|----------|------|------|
| *PIKFYVE* | rs771244880 | chr2:209217490 | c.5828G>A | p.G1943E | 0.00002 | 0.999 | 5.09 | 26.00 |
| *NPHS1* | rs114849139 | chr19:36330456 | c.2869C>G | p.V957L | 0.001 | 0.987 | 4.27 | 14.34 |
| *FPR1* | rs78488639 | chr19:52249959 | c.289G>T | p.L97M | 0.008 | 0.795 | 2.55 | 10.70 |

MAF: minor allele frequency. gnomAD: genome aggregation database. PolyPhen: polymorphism phenotyping. GERP: evidence of evolutionary conservation. CADD: combined annotation dependent depletion.

absent from the unaffected member were filtered and retained. Then we focused on variants in known cataract genes but did not find any disease causative variants in this family. Further filtering based on in-silico prediction of pathogenicity refined the candidates to three genes (*PIKFYVE*, *NPHS1*, and *FPR1*; *Table 2*). We then verified and evaluated these candidate variants in the whole family using Sanger sequencing. Among them, only the *PIKFYVE* variant completely segregated with the disease in the family (*Figure 1A*). As shown in *Figure 2A*, in contrast to the healthy control, the *PIKFYVE* gene in the cataract patients carried a G to A heterozygous substitution in the 39th exon, which leads to the replacement of a highly conserved amino acid glycine (G) by glutamate (E) at position 1943 in the PIPK domain (*Figure 2B and C*). To investigate the possible roles of PIKFYVE in the development of cataracts, we first confirmed its expression in human anterior lens capsules by quantitative RT-PCR (*Figure 2—figure supplement 1A* and *B*). Then to interpret how p.G1943E variant altered the function of PIKFYVE, we determined the protein stability of PIKFYVE$^{WT}$ and PIKFYVE$^{G1943E}$ by Western blot analysis of *pCS2(+)-CMV-PIKFYVE$^{WT}$* and *pCS2(+)-CMV-PIKFYVE$^{G1943E}$* transfected HEK293T cells at 34 and 44 hr post-transfection. As shown in *Figure 2D*, the expression level of PIKFYVE$^{G1943E}$ was comparable with that of PIKFYVE$^{WT}$, suggesting that the p.G1943E variant has little or no effect on the protein stability of PIKFYVE. Recently, a middle-to-low resolution structure of PIKFYVE was resolved by cryo-EM (PDB ID:7K2V) (*Lees et al., 2020*). Using this structure as template, we predicted the structure of the p.G1943E variant form of PIKFYVE PIPK domain. As shown in *Figure 2E and F*, the p.G1943E sits in a linker connecting the N-lobe (gold) and the C-lobe (cyan) of PIPK domain. This loop (red) acts as a hinge between N-lobe and C-lobe (*Figure 2F*) and forms a highly restricted conformation. However, the p.G1943E variant introduces a negatively charged residue to the loop. This negatively charged residue not only changes the surface electrostatic potential of PIPK domain (*Figure 2G*), but also generates charge repulsions between the loop and N-lobe (D1872 and D1871) (*Figure 2F*), thereby potentially affecting the kinase activity of PIKFYVE.

To further strengthen the correlation of PIKFYVE variants with cataracts, we screened another congenital cataract family, 10 sporadic cases with congenital cataract and 200 patients with age-related or complicated cataract. In this experiment, six affected loci from seven age-related or complicated cataract patients were identified. Among them, two variants in exon 36 (c.5399A>C, p.Q1800P) and exon 37 (c.5594C>T, p.A1865V) are located in the PIP kinase domain (*Figure 2—figure supplement 2*, *Figure 2—figure supplement 2—source data 1*). These findings further suggested that genetic variants in PIKFYVE might be causal/risk factors for cataract formation.

## Disruption of PIPK domain of PIKFYVE caused early onset cataract in zebrafish

Bioinformatic analysis suggests that the p.G1943E variant introduces a negatively charged residue in the loop, which potentially affects the kinase activity of PIKFYVE. To directly demonstrate that heterozygosity of the p.G1943E variant caused haploinsufficiency rather than dominant-negative inhibition of PIKFYVE activity in human patients, we generated *pikfyve*-deficient zebrafish mutants using the CRISPR/Cas9-directed gene editing method (*Shankaran et al., 2018*). To maximally mimic the p.G1943E variant in human patients, we designed single-guide RNA (sgRNA) in exon 40 to target the C terminal PIPK domain and screened out the *pikfyve$^{\Delta 8}$* allele, which harbored an 8 bp deletion and a 112 bp insertion in the cDNA of *pikfyve*. As shown in *Figure 3A*, this insertion/deletion in *pikfyve$^{\Delta 8}$* allele introduced a premature stop codon immediately downstream of the insertion site, thereby

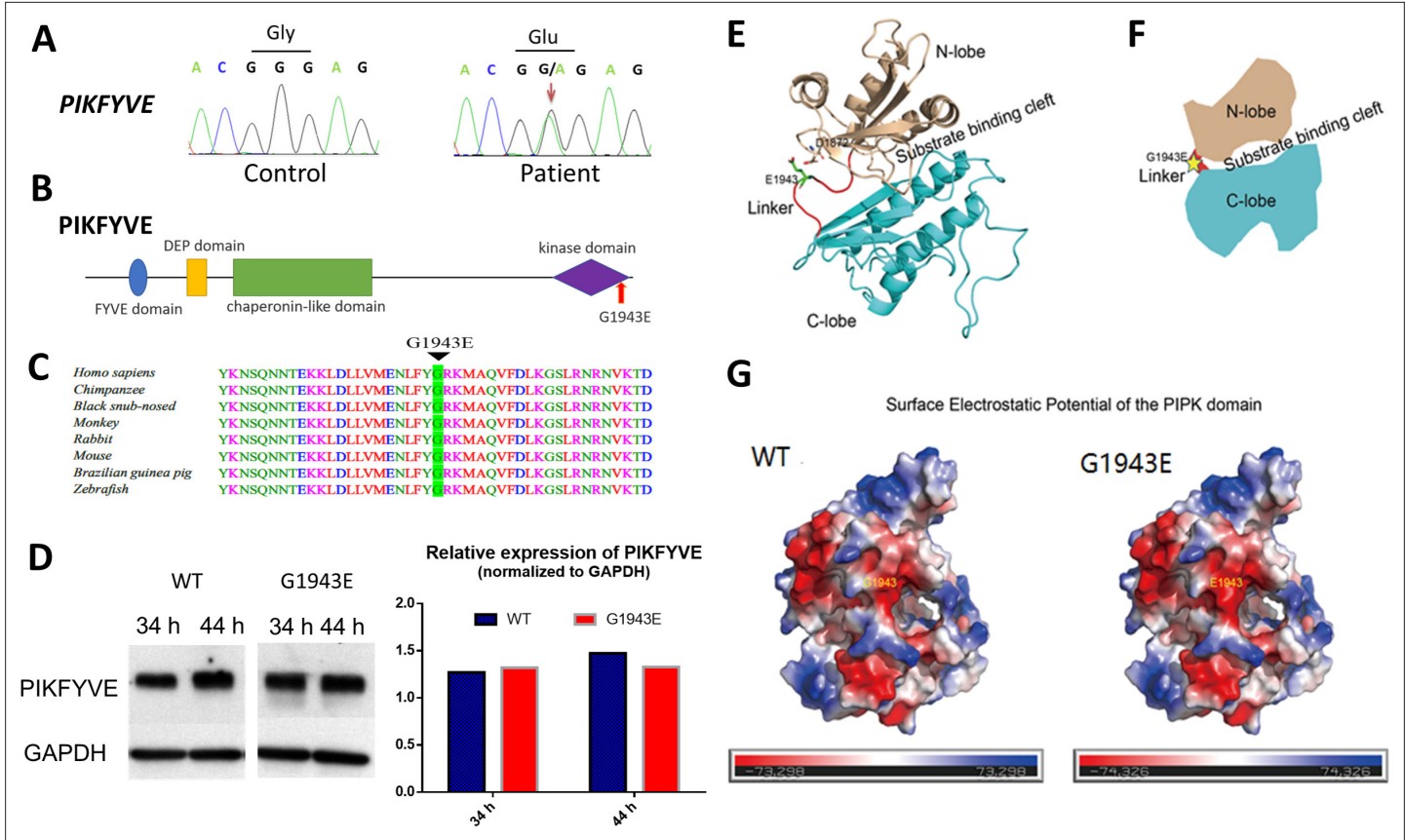

**Figure 2.** The *PIKFYVE* variant identified from the congenital cataract family. (**A**) Sanger sequencing chromatogram showing the cDNA sequences from a healthy control and a cataract patient. The heterozygous c.5828G>A missense variant in the patient is indicated by the red arrow. (**B**) A schematic diagram showing the human PIKFYVE domains. The p.G1943E variant in the PIPK domain is indicated by the red arrow. (**C**) Protein sequence alignment of PIKFYVE orthologs in vertebrates. The black triangle denotes the conserved glycine at position 1943. (**D**) Western blot analysis of PIKFYVE[WT] and PIKFYVE[G1943E] expression in HEK293T cells that were transiently transfected with either *pCS2(+)-CMV-PIKFYVE[WT]* or *pCS2(+)-CMV-PIKFYVE[G1943E]*. The protein levels were normalized by GAPDH expression. Experiments were repeated three times. (**E**) Predicted structure model of the p.G1943E variant form of PIKFYVE PIPK domain generated by the PHYPRE2 server (http://www.sbg.bio.ic.ac.uk/~phyre2/html/). N-lobe, C-lobe, and the hinge linker are shown in gold, cyan, and red, respectively. The variant residue E1943 is shown in sticks and labeled with green. The negatively charged residue D1872 close to E1943 side chain is also shown in sticks. (**F**) A schematic demonstrating the organization of PIKFYVE PIPK domain. N-lobe, C-lobe, and the hinge linker are shown in gold, cyan, and red, respectively. The position of the p.G1943E variant is labeled with a yellow star. (**G**) Surface electrostatic potential comparison of the PIPK domain of PIKFYVE between wild-type (WT) and p.G1943E variant. The electrostatic potentials are presented as heatmaps from red to blue, and the electrostatic potential scales are shown in the lower panel. See *Figure 2—source data 1* for details.

The online version of this article includes the following source data and figure supplement(s) for figure 2:

**Source data 1.** Raw data for intensity of bands in *Figure 2D*.

**Source data 2.** Raw data for the full raw unedited blots in *Figure 2D*.

**Figure supplement 1.** Expression of PIKFYVE in human lens capsule.

**Figure supplement 2.** A schematic diagram showing the distribution of *PIKFYVE* variants.

**Figure supplement 2—source data 1.** Raw data for the clinical manifestation of seven patients with PIKFYVE variants in *Figure 2—figure supplement 2*.

presumably generating a truncated Pikfyve protein that lacks the kinase activity. Phenotypical analysis revealed that *pikfyve[Δ8]* homozygous mutants developed normally before 5 days post-fertilization (dpf), but later showed severe developmental defects and all died by 9 dpf (*Figure 3—figure supplement 1A, B*). Under stereomicroscope, we noticed that compared to siblings (*sib*), the lens of *pikfyve[Δ8]* mutants was less transparent at 5 dpf (*Figure 3B*). To characterize this phenotype in details, we followed up the development of lens in all genotypes at different stages under differential interference contrast (DIC) microscope. Our results showed that bubble-like vacuoles first appeared in the

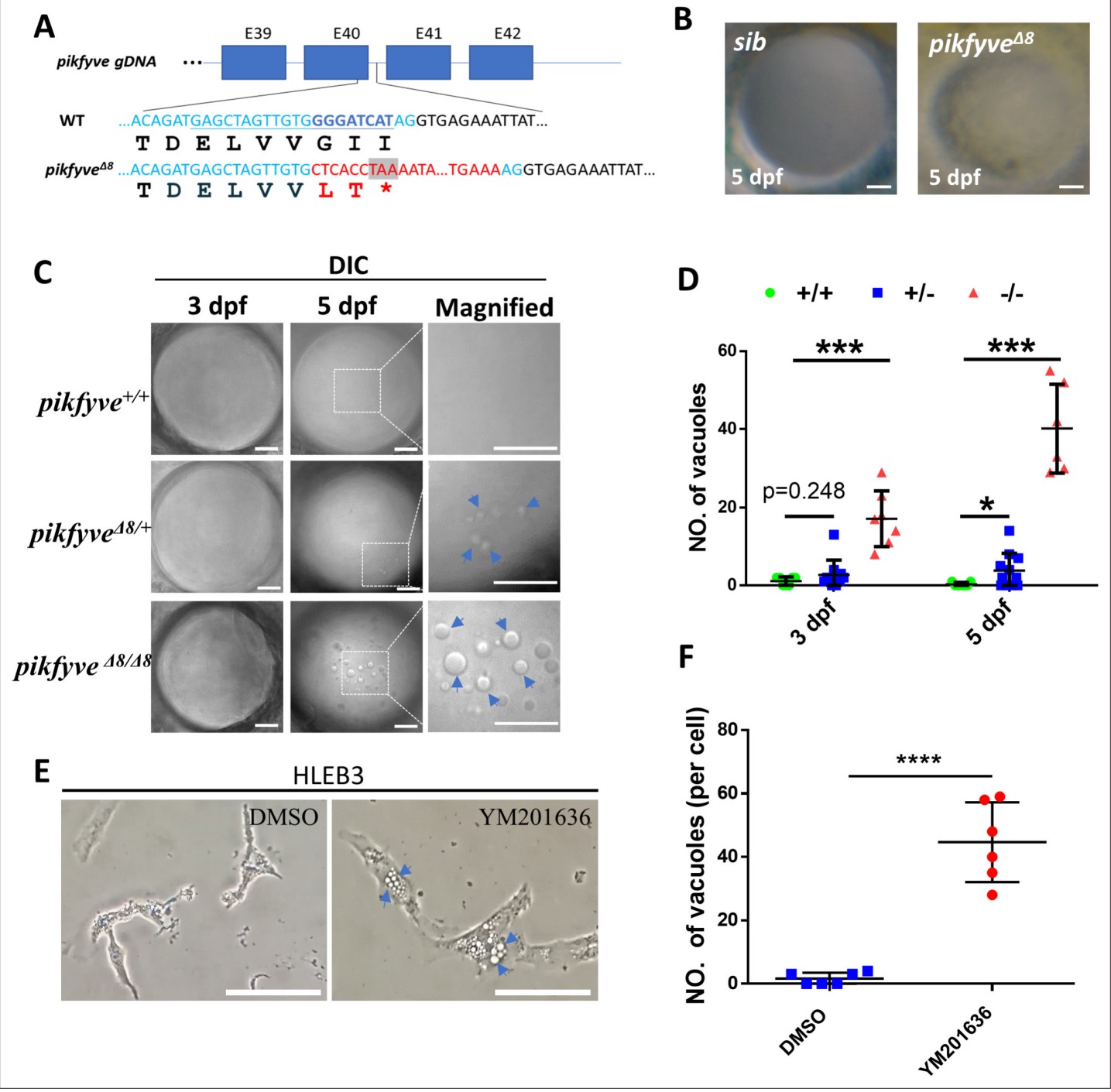

**Figure 3.** Disruption of the PIPK domain of Pikfyve in zebrafish caused early-onset cataract. (**A**) A schematic diagram showing the generated *pikfyve*$^{Δ8}$ mutant allele. The underlined base pairs are the sgRNA target. The deleted base pairs are shown in dark blue while inserted ones are shown in red. The stop codon introduced in the mutant form is shown in the grey box. (**B**) Representative images showing the lens of sibling and *pikfyve*$^{Δ8}$ mutants at 5 dpf. (**C**) Representative differential interference contrast (DIC) images showing the lens of *pikfyve*$^{+/+}$, *pikfyve*$^{+/Δ8}$, and *pikfyve*$^{Δ8/Δ8}$ embryos at 3 dpf and 5 dpf. The scale bars represent 10 μm in (**B**) and (**C**). (**D**) Quantification of vacuole number in the lens of *pikfyve*$^{+/+}$, *pikfyve*$^{+/Δ8}$, and *pikfyve*$^{Δ8/Δ8}$ embryos at 3 dpf (n=7 for *pikfyve*$^{+/+}$; n=10 for *pikfyve*$^{+/Δ8}$; n=7 for *pikfyve*$^{Δ8/Δ8}$) and 5 dpf (n=7 for *pikfyve*$^{+/+}$; n=11 for *pikfyve*$^{+/Δ8}$; n=6 for *pikfyve*$^{Δ8/Δ8}$). (**E**) Representative images of HLEB3 cells treated with DMSO or PIKFYVE inhibitor YM201636 for 4 hr. The scale bars represent 25 μm. (**F**) Quantification of the vacuole numbers in (**E**). ****, p<0.0001, Student's t-test. All experiments were repeated three times. See *Figure 3—source data 1* for details.

The online version of this article includes the following source data and figure supplement(s) for figure 3:

**Source data 1.** Raw data for quantification in *Figure 3D and F*.

*Figure 3 continued on next page*

*Figure 3 continued*

**Figure supplement 1.** Characterization of *pikfyve*-deficient zebrafish mutants.

**Figure supplement 1—source data 1.** Raw data for quantification in *Figure 3—figure supplement 1B*.

**Figure supplement 2.** Ectopic overexpression of *PIKFYVE*$^{G1943E}$ failed to induce cataract defect in zebrafish.

**Figure supplement 3.** The G1943E variant form of *PIKFYVE* is less efficient to rescue the vacuole defect in *pikfyve*-deficient zebrafish mutants.

**Figure supplement 3—source data 1.** Raw data for quantification in *Figure 3—figure supplement 3B, C*.

developing lens of *pikfyve*$^{Δ8}$ homozygous mutants at 3 dpf (*Figure 3C and D*). Strikingly, the number and size of vacuoles both increased drastically by 5 dpf and almost dominated the lens of mutants. Notably, compared to the wild-type (WT) control, *pikfyve*$^{Δ8}$ heterozygous mutants also manifested significantly higher number of vacuoles in their lens (*Figure 3C and D*), suggesting that normal development of lens highly depends on the activity of Pikfyve and heterozygous disruption of Pikfyve is sufficient to cause cataract phenotype. Furthermore, pharmacological inhibition of PIKFYVE activity by inhibitor YM201636 also induced aberrant vacuolation in human lens epithelial cells HLEB3 (*Figure 3E and F*), suggesting that the function of PIKFYVE in the regulation of cataract formation is highly conserved across species and our zebrafish cataract model would potentially be an excellent system for further preclinical study.

In addition to loss of function study of Pikfyve, we also utilized a ubiquitously expressed *ubiquitin* (*ubi*) promoter to generate transgenic lines, that is, *Tg*(*ubi:PIKFYVE*$^{WT}$) and *Tg*(*ubi:PIKFYVE*$^{G1943E}$) to overexpress PIKFYVE$^{WT}$ and PIKFYVE$^{G1943E}$ in zebrafish, respectively (*Figure 3—figure supplement 2A*). As expected, both of these two zebrafish lines showed normal development of the lens during early stages (*Figure 3—figure supplement 2B*), suggesting that indeed haploinsufficiency, rather than dominant-negative inhibition of PIKFYVE activity by the p.G1943E variant caused cataract in human patients. Moreover, the vacuolation phenotype could be partially rescued in both *pikfyve*$^{Δ8}$;*Tg*(*ubi:PIKFYVE*$^{WT}$) and *pikfyve*$^{Δ8}$;*Tg*(*ubi:PIKFYVE*$^{G1943E}$) zebrafish (*Figure 3—figure supplement 3A*). However, while the vacuole numbers decreased to less than 15 in all *pikfyve*$^{Δ8}$;*Tg*(*ubi:PIKFYVE*$^{WT}$) embryos (n=6/6), about one-third of *pikfyve*$^{Δ8}$;*Tg*(*ubi:PIKFYVE*$^{G1943E}$) mutants still harbored more than 15 vacuoles (n=5/17) (*Figure 3—figure supplement 3A*). As whole-mount in situ hybridization (WISH) results revealed that the expression of *PIKFYVE* in *Tg*(*ubi:PIKFYVE*$^{G1943E}$) was at least as strong as or even stronger than that of *PIKFYVE*$^{WT}$ (*Figure 3—figure supplement 3B*). These results indicated that the rescue effect of PIKFYVE$^{G1943E}$ is less efficient than that of PIKFYVE$^{WT}$.

## Detailed characterization of cataract phenotypes in *Pikfyve*$^{Δ8}$ mutants

To delineate the details of cataract phenotype in *pikfyve*$^{Δ8}$ mutants, we crossed *pikfyve*$^{Δ8}$ mutant with the reporter transgenic line *Tg*(*cryaa:DsRed*) (*Nguyen-Chi et al., 2014*), in which expression of DsRed was controlled by promoter of the crystalline gene *cryaa*, the major constitutive components of lens fibers (*Runkle et al., 2002*). Intriguingly, in comparison with evenly distributed DsRed signals in the lens of siblings, vacuoles in *pikfyve*$^{Δ8}$ mutants almost occupied the surface of the lens and all of them were DsRed negative (*Figure 4A*), indicating that these vacuoles did not contain lens fibers. Meanwhile, hematoxylin-eosin (HE) staining, together with ZL-1 antibody and DAPI co-staining revealed that while disruption of Pikfyve seemingly had no effects on the enucleation process of lens, lens fibers in *pikfyve*$^{Δ8}$ mutants were less organized than those in siblings (*Figure 4B and C*). To further characterize the vacuoles and lens structure in high resolution, we utilized transmission electron microscope (TEM) to visualize the ultrastructure of lens in both siblings and *pikfyve*$^{Δ8}$ mutants. We could detect large vacuoles in the mutant lens at 3 dpf and 5 dpf, while only several tiny vacuoles were observed in the sibling lens (*Figure 4D*). Lens fiber cells of siblings were mildly edema, and their nuclei were oval with uniform chromatin. The lens fibers (white arrows) were arranged neatly and tightly. Furthermore, mitochondria (black triangles) of siblings were slightly swollen without vacuoles. In contrast, lens fiber cells of the *pikfyve*$^{Δ8}$ mutants were obviously edematous and the nuclei were irregularly shaped. Besides, the arrangement of lens fibers was loose and deformed and lipid droplets were formed in *pikfyve*$^{Δ8}$ mutants. Compared to the siblings, mitochondria (black triangles) of *pikfyve*$^{Δ8}$ mutants were obviously swollen and enlarged. Mitochondrial crests were reduced, and most of them were aberrantly vacuolated. In addition, autophagic lysosomes also appeared swollen and vacuolated in *pikfyve*$^{Δ8}$ mutants.

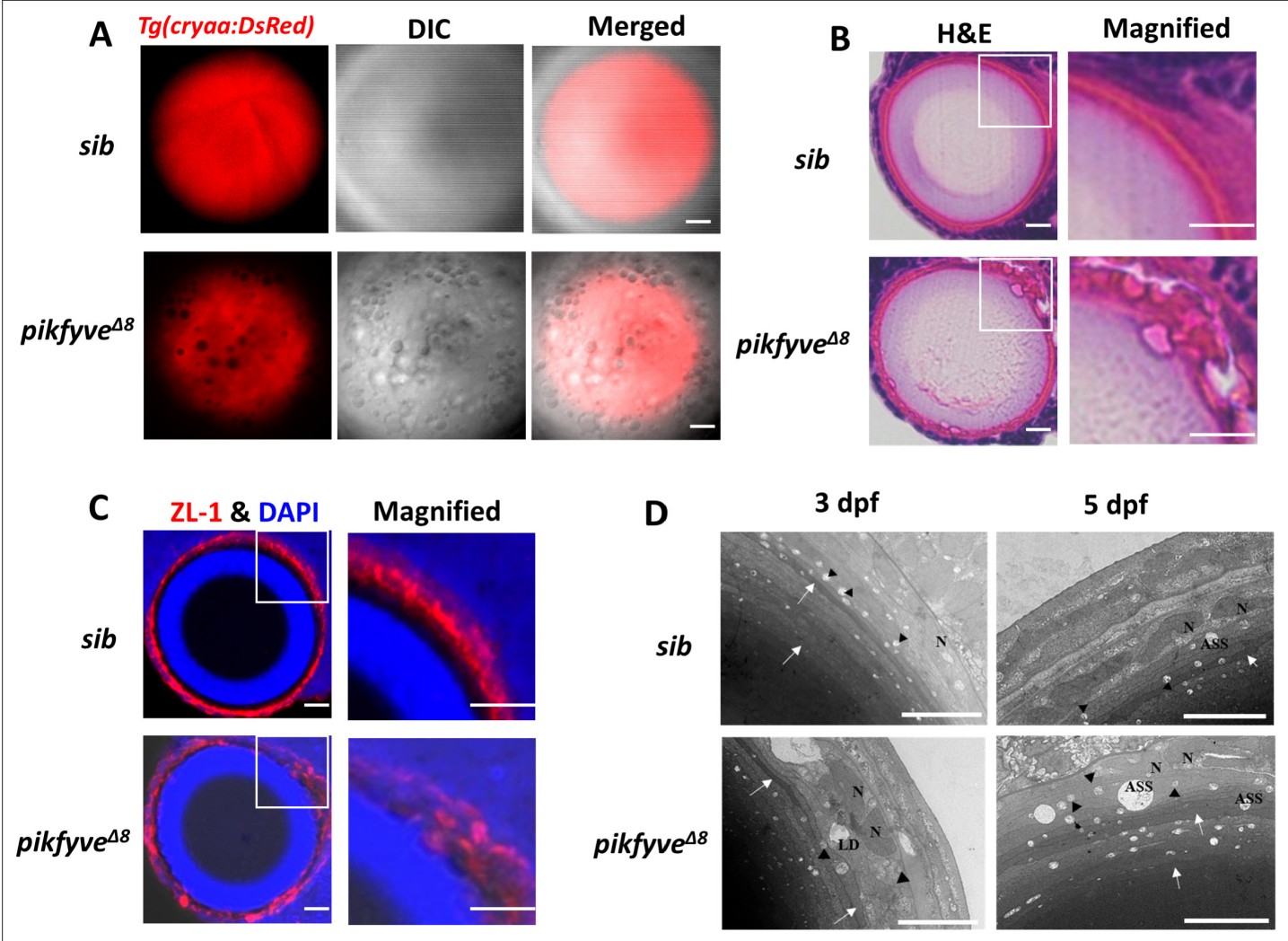

**Figure 4.** Detailed characterization of cataract phenotypes in *pikfyve*^Δ8^ mutants. (**A**) Confocal imaging of the lens of 5-dpf siblings and *pikfyve*^Δ8^ mutants in *Tg*(*cryaa:DsRed*) transgenic background. (**B**) Hematoxylin-eosin (HE) staining of 5-dpf siblings and *pikfyve*^Δ8^ mutant zebrafish lens after cryostat section. (**C**) ZL-1 antibody and DAPI staining of 5-dpf siblings and *pikfyve*^Δ8^ mutant zebrafish lens. (**D**) Transmission electron microscope (TEM) images of the lens of siblings and *pikfyve*^Δ8^ mutants at 3 dpf and 5 dpf. ASS, autophagy lysosome; LD, lipid droplet; N, nucleus. All results were confirmed in three different individuals. All the scale bars represent 10 µm.

## Vacuoles in *Pikfyve*^Δ8^ mutants were amphisomes

To define the nature of vacuoles in the lens of *pikfyve*^Δ8^ mutants, we conducted time-lapse imaging to monitor their behaviors during zebrafish development. Our results showed that vacuoles in *pikfyve*^Δ8^ mutants were highly dynamic and small vacuoles were frequently found to be fused with each other to form larger ones (*Figure 5A*, white arrows). This feature, together with previous findings showing that PIKFYVE was an essential regulator of endomembrane homeostasis (*Hasegawa et al., 2017*), prompted us to further investigate whether vacuoles in *pikfyve*^Δ8^ mutants were virtually endocytic vesicles. To probe this issue, we generated fusion mRNAs encoding GFP and the markers of endocytic vesicles (i.e., the small GTPase Rab5c, Rab7, and Rab11a) (*Langemeyer et al., 2018*) and injected them into zebrafish embryos to specifically label early, late, and recycling endosomes in *pikfyve*^Δ8^ mutants. Our results showed that in *pikfyve*^Δ8^ mutants, almost all vacuoles were positive for the late endosome maker Rab7-GFP (*Figure 5C*, white arrows). By contrast, the early endosome marker Rab5c-GFP and recycling marker Rab11a-GFP showed no co-localization with vacuoles (*Figure 5B-D*). As autophagy was also closely related to the organelle membrane system and had been shown to be regulated by PIKFYVE (*Vicinanza et al., 2015*), we therefore checked the status of autophagosomes

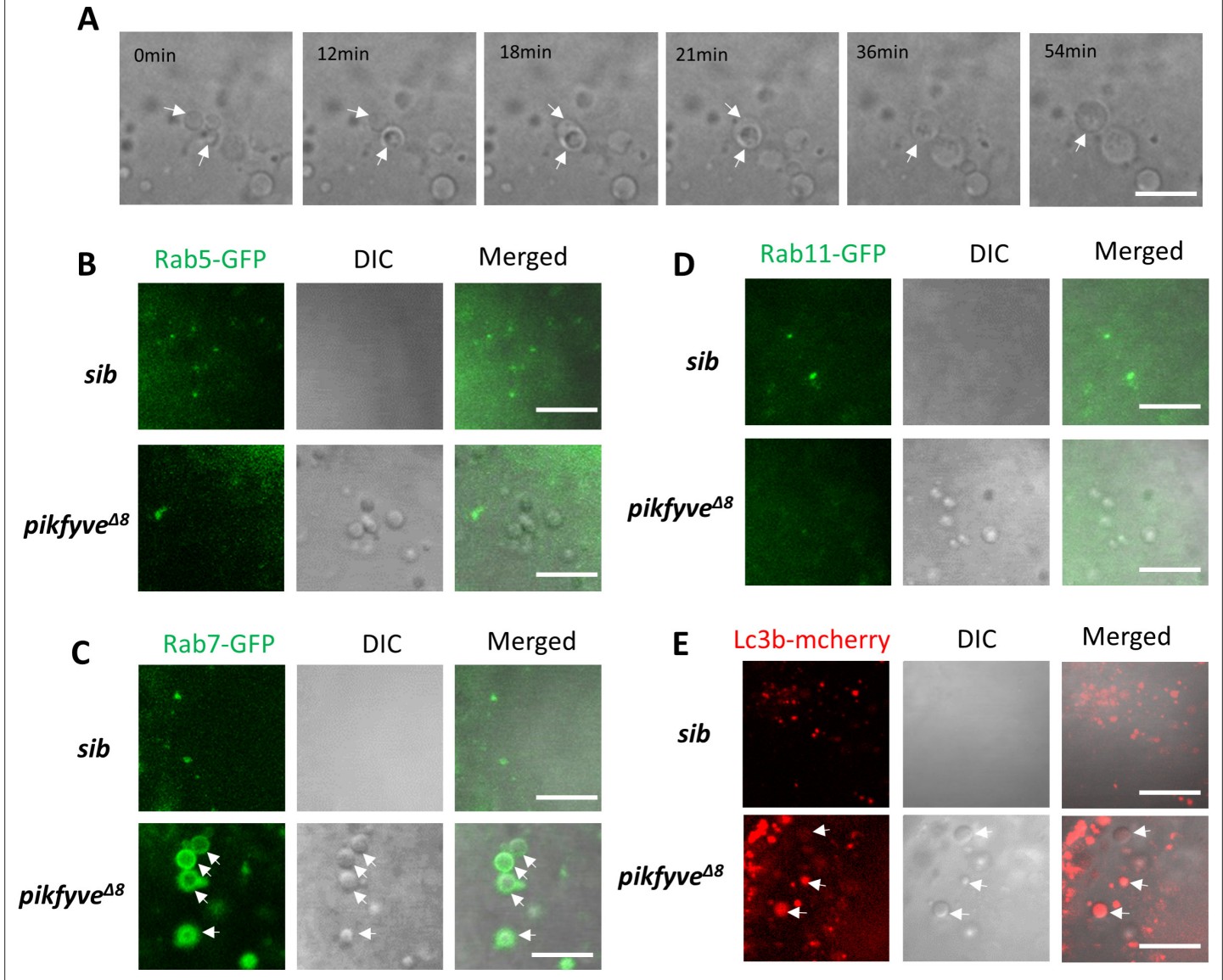

**Figure 5.** Characterization of vacuoles in *pikfyve*^Δ8 mutants. (**A**) Time-lapse imaging indicating the dynamic changes of vacuole formation in the lens of 4-dpf *pikfyve*^Δ8 mutants. White arrows indicate the fusion process of two small vacuoles. (**B**) Representative images showing the lens of 3.5-dpf siblings and *pikfyve*^Δ8 mutants injected with *gfp-rab5c* mRNA. (**C**) Representative images showing lens of 3.5-dpf siblings and *pikfyve*^Δ8 mutants injected with *gfp-rab7* mRNA. (**D**) Representative images showing lens of 3.5-dpf siblings and *pikfyve*^Δ8 mutants injected with *gfp-rab11a* mRNA. (**E**) Representative images showing lens of 3.5-dpf siblings and *pikfyve*^Δ8 mutants injected with *mcherry-lc3b* mRNA. All experiments were repeated three times. All the scale bars represent 10 μm.

The online version of this article includes the following figure supplement(s) for figure 5:

**Figure supplement 1.** Characterization of lysosomes in microglia and lens of *pikfyve*^Δ8 mutant zebrafish.

in *pikfyve*^Δ8 mutants by injecting *lc3b-mcherry* fusion mRNA. As shown in *Figure 5* and a proportion of vacuoles in *pikfyve*^Δ8 mutants were also positive for the autophagosome marker Lc3b. Taken together, these data implied that vacuoles in *pikfyve*^Δ8 mutants were amphisomes formed by the fusion of autophagosomes and late endosomes.

## Baf-A1 partially rescued the vacuole defect in the lens of *Pikfyve*^Δ8 mutant zebrafish

Baf-A1 is a specific inhibitor of V-ATPase. Previous work has shown that the vacuole phenotype induced by PIKFYVE deficiency in COS-7 cells could be rescued by Baf-A1 (*Compton et al., 2016*). We thus

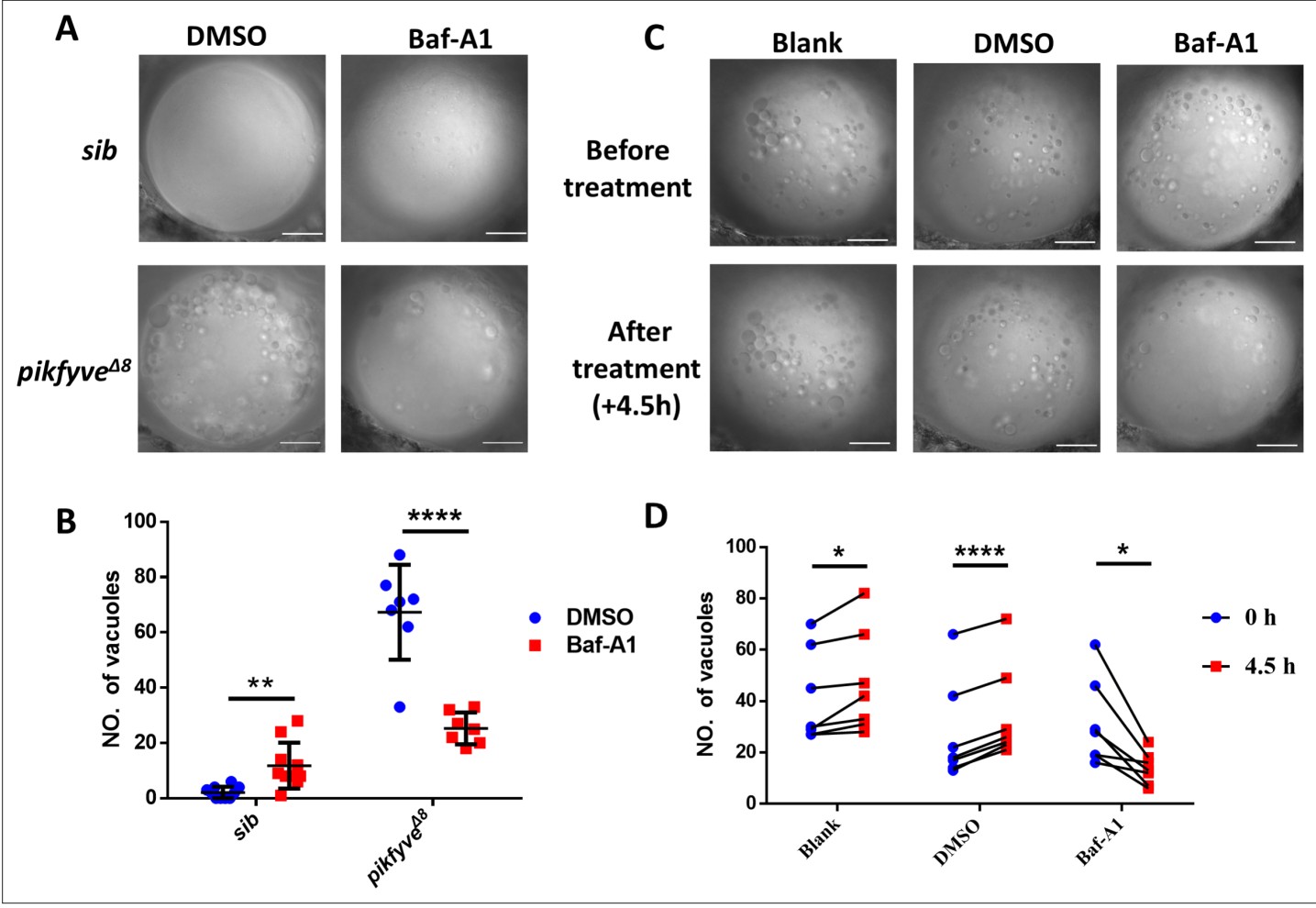

**Figure 6.** Baf-A1 partially rescued the vacuole defect in the lens of *pikfyve^{Δ8}* mutant zebrafish. (**A**) Representative confocal images of the lens of 4-dpf *pikfyve^{Δ8}* mutants treated with DMSO or Baf-A1 for 4.5 hr. (**B**) Quantification of the vacuole numbers in the lens of 4-dpf siblings and *pikfyve^{Δ8}* mutant embryos treated with DMSO or Baf-A1 (n=10 for sibling groups; n=7 for mutant groups). (**C**) Confocal images of the lens of 4-dpf *pikfyve^{Δ8}* mutants with no treatment or after 4.5 hr treatment with DMSO or Baf-A1. (**D**) Quantification of the vacuole numbers in (**C**) (n=6 for each group). All experiments were repeated three times. All the scale bars represent 20 μm. *, $p<0.05$; **, $p<0.01$; ***, $p<0.001$; ****, $p<0.0001$, Student's t-test. See **Figure 6—source data 1** for details.

The online version of this article includes the following source data and figure supplement(s) for figure 6:

**Source data 1.** Raw data for quantification in **Figure 6B and D**.

**Figure supplement 1.** Overexpression of *trpml1* failed to rescue the lens defects in *pikfyve ^{Δ8}* mutants.

**Figure supplement 1—source data 1.** Raw data for quantification in **Figure 6—figure supplement 1B**.

investigated whether the vacuolation defect in the lens of *pikfyve^{Δ8}* mutant zebrafish could also be attenuated by Baf-A1. Indeed, we found that vacuole number in the lens of Baf-A1-treated *pikfyve^{Δ8}* mutants was significantly lower than that in the control group treated with dimethyl sulfoxide (DMSO) (**Figure 6A and B**). To further validate that Baf-A1 could directly alleviate cataract defect, rather than just delay the phenotype, we imaged the lens of the same mutant zebrafish before and after Baf-A1 treatment. As shown in **Figure 6C and D**, while vacuole numbers of all the mutant embryos showed a slight increase after DMSO treatment or without treatment, we observed significant decrease of vacuole numbers in all *pikfyve^{Δ8}* mutants. On the other hand, a previous study also showed over-expression of transient receptor potential mucolipin 1 (TRPML1) could partially rescue the vacuole phenotype in PIKfyve-deficient macrophages (**Krishna et al., 2016**). However, in our system, ectopic expression of Trpml1 by mRNA injection failed to rescue vacuolation phenotype in *pikfyve^{Δ8}* mutants (**Figure 6—figure supplement 1**). Thus, V-ATPase rather than Trpml1 might be a key component for

the formation or maintenance of the large vacuoles in the lens of *pikfyve*-deficient mutants, and inhibition of V-ATPase by Baf-A1 could directly alleviate the cataract phenotype.

## Discussion

In this study, we identified a missense variant (p.G1943E) in *PIKFYVE* responsible for congenital cataract in a Chinese Korean family (*Figures 1 and 2*). This variant is very rare in the general population. In the gnomAD database, only four individuals (including three East Asians and one Latino/Admixed American) carry the p.G1943E variant, representing an extremely low allele frequency of 0.00002 (*Table 2*). Consistent with previous findings (*Berry et al., 2020b*), the degree and morphology of lens opacity in the cataract family were phenotypically heterogeneous (*Table 1*). While most of the patients in this family developed cataract and vision loss in their childhood, some cases were diagnosed quite late. Meanwhile, the majority of patients had nuclear pulverulent cataract, while the others had nuclear Y-sutural cataract or peripheral cortical punctate cataract (*Table 1*). This heterogeneity in clinical manifestations might be ascribed to interactions between genetic and environmental factors during lens development (*Berry et al., 2020b*).

Variants in *PIKFYVE* have been reported to be associated with CFD (*Gee et al., 2015*; *Kawasaki et al., 2012*; *Kotoulas et al., 2011*; *Li et al., 2005*). However, except that two of the CFD patients showed cataract formation at 66 and 58 years old, respectively, none of these *PIKFYVE* variants in these studies caused congenital cataract (*Kotoulas et al., 2011*). Interestingly, we also did not observe any corneal defect in patients with congenital cataract in this study. It is noted that all CFD-related mutations in *PIKFYVE*, either homozygous or heterozygous, are distributed in two regions (i.e., amino acids 667–843 and 1490–1538), corresponding to cytosolic chaperone CCTγ apical domain-like motif and SPEC domains, respectively. By contrast, the p.G1943E variant identified in this study is located in the C-terminal PIPK domain. Based on these findings, we hypothesized that different domains of PIKFYVE might exert different functions by binding with different partners. This hypothesis is further supported by other evidences. First, the predicted protein structure suggested that, instead of gross disruption of PIKFYVE protein structure, the p.G1943E variant only mildly altered the 3-D conformation of PIPK domain by changing its surface electrostatic potential and generating charge repulsions between the loop and N-lobe (*Figure 2G and F*). Therefore, it is reasonable to argue that this mutation in PIPK domain may specifically alter its kinase activity or change the binding surface with other proteins, while keep the N-terminal cytosolic chaperone CCTγ apical domain-like motif and SPEC domains functional intact. Second, in our *pikfyve*^Δ8 zebrafish mutants, while the C-terminal PIPK domain was truncated in Pikfyve^Δ8 protein, the N-terminal cytosolic chaperone CCTγ apical domain-like motif and SPEC domains were not affected. Accordingly, we did not observe any cornea defect in the zebrafish mutants either.

Inhibition of PIKFYVE in COS-7 cells had been shown to induce large vacuoles through promoting the enlargement of both early and late endosomes (*Ikonomov et al., 2006*; *Rutherford et al., 2006*). Further studies revealed that the Ca^{2+} releasing channel endolysosome-localized mucolipin TRPML1 acts downstream of PIKFYVE to trigger membrane fusion/fission process (*Dong et al., 2010*) or promote lysosome/phagosome maturation (*Dayam et al., 2015*; *Kim et al., 2014*). Interestingly, while overexpression of TRPML1 partially alleviated the vacuole phenotype in PIKFYVE-deficient macrophages (*Krishna et al., 2016*), the rescue effect was not observed in the lens of *pikfyve*^Δ8 zebrafish mutants (*Figure 6—figure supplement 1*), suggesting that PIKFYVE might function in a context-dependent manner. Consistent with this idea, we also noted that vacuoles in microglia (the brain-resident macrophages) and lens cells of *pikfyve*^Δ8 mutants were differently stained by lysosome marker (*Figure 5—figure supplement 1*). On the other hand, vacuole formation in PIKFYVE-deficient COS-7 cells and macrophages could also be inhibited by a drug called Baf-A1 (*Compton et al., 2016*; *Isobe et al., 2019*). In this study, we found that Baf-A1 could partially rescue the vacuole defect in the lens of *pikfyve*^Δ8 mutant zebrafish (*Figure 6A–D*). Baf-A1 is a macrolide antibiotic that inhibits V-ATPase, the ATP-dependent proton pump located on the membrane of organelles. The cellular acidification process mediated by V-ATPase may affect many basic biological processes, including membrane trafficking (in particular endosome maturation and fusion between autophagosomes and lysosomes; *Hammond et al., 1998*; *Yamamoto et al., 1998*), protein degradation, and autophagy (*Bowman et al., 1988*; *Yamamoto et al., 1998*; *Yoshimori et al., 1991*). Interestingly, several works in yeast also identified mutations in V-ATPase that did not affect proton pump function, but indeed caused

defects in vacuole fusion (*Strasser et al., 2011*). Moreover, another study in *Drosophila* demonstrated that inhibition of vesicle fusion by Baf-A1 did not depend on V-ATPases, but relied on $Ca^{2+}$ sarco/endoplasmic reticulum $Ca^{2+}$-ATPase (SERCA) pump, the secondary target of Baf-A1 (*Mauvezin et al., 2015*). Therefore, the target of Baf-A1 and underlying mechanisms still need to be further investigated. These mechanisms will finally contribute to the potential clinical application of the drug in the treatment of congenital cataract caused by PIKFYVE deficiency.

# Materials and methods

## Key resources table

| Reagent type (species) or resource | Designation | Source or reference | Identifiers | Additional information |
|---|---|---|---|---|
| Cell line (*Homo sapiens*) | HEK293T | Laboratory cell bank of Shenzhen PKU-HKUST Medical Center | | Initially ordered from ATCC by colleagues in HKUST |
| Cell line (*H. sapiens*) | HLEB3 | ATCC | ATCC CRL-11421, RRID:CVCL_6367 | |
| Strain, strain background (*Danio rerio*) | *pikfyve*$^{\Delta 8}$ mutant | This paper | | Maintained in Shenzhen PKU-HKUST Medical Center |
| Strain, strain background (*D. rerio*) | *Tg(ubi:PIKFYVE$^{WT}$)* | This paper | | Maintained in Shenzhen PKU-HKUST Medical Center |
| Strain, strain background (*D. rerio*) | *Tg(ubi:PIKFYVE$^{G1943E}$)* | This paper | | Maintained in Shenzhen PKU-HKUST Medical Center |
| Strain, strain background (*D. rerio*) | *Tg(cryaa:DsRed;il-1b:GFP-F)* | doi.10.1242/dmm.014498 | | |
| Chemical compound, drug | YM201636 | Selleck, China | S1219 | 800 nM for 4 hr |
| Chemical compound, drug | bafilomycin A1 | MedChemExpress, NJ | HY-100558 | 1 μM for 4.5 hr |
| Antibody | Anti-PIKFYVE antibody (rabbit polyclonal) | Abcam, UK | Cat# ab137907 | WB (1:1000) |
| Antibody | Anti-GAPDH antibody (rabbit polyclonal) | Abcam, UK | Cat# ab9485, RRID:AB_307275 | WB (1:1000) |
| Antibody | Anti-Rabbit IgG H&L (goat polyclonal) | Abcam, UK | Cat# ab205718, RRID:AB_2819160 | WB (1:5000) |
| Antibody | Anti-Lens Fiber Cell Marker antibody [ZL-1] (mouse monoclonal) | Abcam, UK | Cat# ab185979 | IF (1:100) |
| Antibody | Anti-Mouse-555 secondary antibody (donkey polyclonal) | Thermo Fisher Scientific, USA | Cat# A-31570, RRID:AB_2536180 | IF (1:400) |
| Sequence-based reagent (*H. sapiens*) | Human PIKFYVE genotyping fwd | This paper | PCR primer | TTTTGACCTTCTCTTGATTAGAGG |
| Sequence-based reagent (*H. sapiens*) | Human PIKFYVE genotyping rev | This paper | PCR primer | AAATATGGCCTAGTAACCAAAGTTAAA |
| Sequence-based reagent (*H. sapiens*) | Human PIKFYVE cDNA cloning fwd | This paper | PCR primer | ATGGCCACAGATGATAAGAC |

*Continued on next page*

*Continued*

| Reagent type (species) or resource | Designation | Source or reference | Identifiers | Additional information |
|---|---|---|---|---|
| Sequence-based reagent (*H. sapiens*) | Human PIKFYVE cDNA cloning rev | This paper | PCR primer | TCAGCAATTCAGACCCAAGCCTG |
| Sequence-based reagent (*H. sapiens*) | Human PIKFYVE qPCR fwd | This paper | PCR primer | CGTCCCCAACACTGGACTCTGC |
| Sequence-based reagent (*H. sapiens*) | Human PIKFYVE qPCR rev | This paper | PCR primer | CCCTGGCCTCCTTCTGCTCTCTC |
| Sequence-based reagent (*H. sapiens*) | Human GAPDH qPCR fwd | This paper | PCR primer | CGAGATCCCTCCAAAATCAA |
| Sequence-based reagent (*H. sapiens*) | Human GAPDH qPCR rev | This paper | PCR primer | GTCTTCTGGGTGGCAGTGAT |
| Sequence-based reagent (*D. rerio*) | Zebrafish *pikfyve* genotyping fwd | This paper | PCR primer | GAGAACCTGCTCAAACTGGTGC |
| Sequence-based reagent (*D. rerio*) | Zebrafish *pikfyve* genotyping rev | This paper | PCR primer | AGATTTGACCACCATCTCCAGC |

## Pedigree and patients

A four-generation pedigree consisting of 31 family members (*Figure 1A*) was recruited from the eye clinic of Shenzhen Eye Hospital, Shenzhen, Guangdong, China. All the living family members underwent a complete ocular examination. Twelve individuals in this family were diagnosed with congenital cataract (*Figure 1B*, *Table 1*), while the other 19 individuals were unaffected. The diagnosis of congenital cataract was based on (1) lens opacity detected at birth or during the first decade of life; and (2) no known other causes (e.g., trauma, iatrogenic, or inflammatory disease) (*Berry et al., 2020a*). Meanwhile, we also recruited another congenital cataract family, 10 sporadic cases with congenital cataract and 200 patients with age-related cataract or complicated cataract. The ethical protocol in this study was approved by the Independent Ethics Committee of Shenzhen Eye Hospital, in accordance with the tenets of the Declaration of Helsinki (1975). Written informed consent and consent to publish were received from all study participants prior to obtaining the human tissue samples. Samples were de-identified by removing the identifiable individual information such as name and identity number before researching.

## Identification of genetic variants in *PIKFYVE*

WES and Sanger sequencing were used in this study to identify the p.G1943E genetic variant in the congenital cataract family. In brief, DNA was extracted from blood or hair samples from cataract patients. Exome capture was performed using a SureSelect Human All Exon V6 Capture Kit (Agilent Technologies, Santa Clara, CA). WES was performed on four selected family members (II-1, II-5, III-9, and IV-5) of Chinese Korean family. Samples were sequenced on a HiSeq 4000 next-generation sequencing system (Illumina, San Diego, CA). Sequence reads were aligned to the human reference genome (hg19) using the Burrows-Wheeler Aligner (BWA) (*Li and Durbin, 2009*). Sequence variants were called using the Genome Analysis Toolkit (GATK) with the Best Practices for SNP and Indel discovery in germline DNA (*McKenna et al., 2010*). The variants were annotated using Annotate Variation (ANNOVAR) (*Wang et al., 2010*) and then filtered based on the following criteria: (1) nonsynonymous SNPs or indels in the exon region or the splice site region; (2) novel or minor allele frequency <1% in gnomAD; (3) PolyPhen score >0.2; (4) genomic evolutionary rate profiling (GERP) score >2.5; and (5) combined annotation dependent depletion score (CADD) >10. Polymerase chain reaction (PCR) and Sanger sequencing were used to verify the identified mutations from WES and screen the other family members for the mutations. The primer sequences used in PCR and Sanger

sequencing were: (1) forward primer: 5'-TTTTGACCTTCTCTTGATTAGAGG-3' and (2) reverse primer: 5'-AAATATGGCCTAGTAACCAAAGTTAAA-3'.

For another congenital cataract family and 10 sporadic cases with congenital cataract, gene panel testing of ophthalmology (testing of known pathological genes in cataract and *PIKFYVE*) were used, while for 200 patients with age-related cataract or complicated cataract, fast-target sequencing of *PIKFYVE* exons were utilized to screen for other potential pathological variants in *PIKFYVE*. The variants were filtered based on the following criteria: (1) sorting intolerant from tolerant (SIFT) score ≤ 0.05; (2) deleterious annotation of genetic variants using neural networks (DANN) score >0.93; (3) variant effect scoring tool (VEST) score from 0 to 1, the higher score, the more likely mutation might cause a functional change; (4) ClinPred score >0.5; and (5) MutationTaster score >0.5.

## Cell culture

The HLEB3 cells (ATCC CRL-11421) were ordered from ATCC. HEK293T cells were obtained from the Laboratory cell bank of Shenzhen PKU-HKUST Medical Center, which were initially ordered from ATCC by colleagues in HKUST. Cell culture was performed according to the instructions (*Andley et al., 1994*). The mycoplasma contamination was prevented by examining the integrity of cell nucleus via DAPI staining.

## YM201636 treatment

The HLEB3 cells were plated into six-well plates and grown for 4 days. The cells were then treated with YM201636 (Selleck, China) at a working concentration of 800 nM for 4 hr before imaging. Cells only treated with DMSO were used as control.

## Quantitative RT-PCR

Anterior lens capsules from three individuals with cataract were obtained during the surgery. Total RNA was extracted with FastPure Cell/Tissue Total RNA Isolation Kit (RC112-01, Vazyme, China) and subjected to reverse transcription with SuperScript III First-Strand Synthesis System (18080051, Thermo Fisher Scientific, USA). Quantitative RT-PCR was conducted with iTaq Universal SYBR Green Supermix (1725124, Bio-Rad, USA) in CFX96 Real-Time PCR Detection System.

## Protein detection by western blot

Transfection of the HEK293T cells was performed with Lipofectamine RNAiMAX Reagent (13778030, Thermo Fisher Scientific, USA) according to the manufacturer's instructions. Cells were transfected with 2 µg *pCS2(+)-CMV-PIKFYVE^WT^* or *pCS2(+)-CMV-PIKFYVE^G1943E^* and harvested after 34 and 44 hr for Western blot analysis. Expression of the housekeeping gene *GAPDH* was used as the loading control. The ImageJ software (National Institutes of Health, Bethesda, MD, https://imagej.nih.gov/ij/) was used to measure the intensity of the bands and quantify the expression of PIKFYVE protein. The anti-PIKFYVE (ab137907, Abcam, Cambridge, UK), anti-GAPDH (ab9485, Abcam) primary antibodies, and goat anti-rabbit (ab205718, Abcam) secondary antibodies were used in this study.

## Zebrafish husbandry

Zebrafish were raised and maintained according to the standard protocols (*Westerfield, 1993*). The WT AB strain, *pikfyve^Δ8^*mutants, and transgenic lines including *Tg(cryaa:DsRed;il-1b:GFP-F)* (*Nguyen-Chi et al., 2014*), *Tg(ubi:PIKFYVE^WT^)*, and *Tg(ubi:PIKFYVE^G1943E^)* were used in this study.

## Generation of zebrafish *Pikfyve* mutant

The sgRNA was designed with the online tool Crisprscan (*Vejnar et al., 2016*). The sgRNA was synthesized by in vitro transcription with the MEGAshortscript T7 Transcription Kit (AM1354, Ambion, Austin, TX) and purified by the MEGAclear Kit (AM1908, Ambion). The sgRNA, together with the Cas9 protein (EnGen Cas9 NLS, *S. pyogenes*, M0646M, NEB, Ipswich, MA), was injected into zebrafish embryos at one-cell stage. The final concentration was 100 ng/µl for each sgRNA and 800 ng/µl for Cas9 protein. Stable mutant line was screened by the T7 endonuclease 1 (T7E1) digestion and Sanger sequencing.

## Generation of transgenic zebrafish lines

The coding sequence of WT and G1943E variant form of human *PIKFYVE* was placed downstream of the *ubi* promoter (*Mosimann et al., 2011*), which were cloned to a modified PBSK vector containing

two arms of Tol2 elements (*Kawakami et al., 2000*). The plasmid and transposon mRNA were then injected into embryos at one-cell stage. Injected embryos were raised into adulthood and screened for germline transmission by PCR and WISH (*Zhang and Liu, 2013*) of F1 embryos.

## Histology

Zebrafish embryos were fixed in 4% paraformaldehyde at 5 dpf and then dehydrated in 30% sucrose. The whole embryos were mounted in optimal cutting temperature compound and frozen in –80°C. Cryo-section was performed at 10 μM intervals in transverse planes from the head. HE staining and antibody staining were conducted according to the standard protocols. ZL-1 antibody (ab185979, Abcam) was used in this study. TEM analysis of the lens was completed by Servicebio Technology (Wuhan, China).

## Imaging of zebrafish lens

Images were taken under ZEISS LSM980 Confocal Laser Scanning Microscope (Carl Zeiss Microscopy, NY). The 63× objective (Plan-Apochromat 63×/1.40 Oil DIC M27) was used in this study. For time-lapse imaging experiment, 4-dpf zebrafish was mounted in 1.0% low-melting agarose with 0.02% tricaine. The lens was directly imaged under ZEISS Celldiscoverer 7 microscope (Carl Zeiss Microscopy). A 50× water lens (Plan-Apochromat 50×/1.2) was used and images were taken every 3 min. Images were analyzed by ImageJ (National Institutes of Health).

## Detection of endosomes and autophagosomes in zebrafish lens

Coding sequences of *rab5c*, *rab7*, or *rab11a* were fused with *GFP* sequence and cloned into the pCS2+ vector. mRNAs were synthesized in vitro using the mMESSAGE mMACHINE SP6 Kit (AM1908, Ambion). About 1–2 nl mRNA (100 ng/μl) was injected into zebrafish embryos at one-cell stage. At 3.5 dpf, embryos were mounted in 1% low-melting agarose and imaged under the confocal microscope.

## Baf-A1 treatment

Baf-A1 powder (HY-100558, MedChemExpress, NJ) was dissolved to the concentration of 1 mM in DMSO for stock, and diluted to 1 μM in egg water before use. Zebrafish embryos were treated with either 1‰ DMSO or 1 μM Baf-A1 solution at 28.5°C for 4.5 hr and washed out by egg water before confocal imaging.

## Statistical analysis

Two-tailed Student's t-tests were used for comparisons between two groups. Data were represented as mean ± SD. Statistical significance was shown as n.s., $p > 0.05$, *$p < 0.05$, ***$p < 0.001$, and ****$p < 0.0001$.

## Acknowledgements

The authors would like to thank all family members who participated in this study. The authors thank Dr. Georges Lutfalla from Institut Pasteur for providing *Tg*(*cryaa:DsRed;il-1b:GFP-F*) zebrafish. The authors also thank Dr. Keyu Chen from Shenzhen Bay Laboratory for the structure analysis on PIKFYVE. This study was supported by Science, Technology and Innovation Commission of Shenzhen Municipality Grants (GJHZ 20180420180937076 and JCYJ20180228164400218) and Sanming Project of Medicine in Shenzhen Grant (SZSM201812090).

## Additional information

### Funding

| Funder | Grant reference number | Author |
|---|---|---|
| Science, Technology and Innovation Commission of Shenzhen Municipality | GJHZ20180420180937076 | Jun Zhao |

| Funder | Grant reference number | Author |
|---|---|---|
| Science, Technology and Innovation Commission of Shenzhen Municipality | JCYJ20180228164400218 | Jun Zhao |
| Sanming Project of Medicine in Shenzhen | SZSM201812090 | Jun Zhao |

The funders had no role in study design, data collection and interpretation, or the decision to submit the work for publication.

## Author contributions

Shaoyi Mei, Conceptualization, Data curation, Investigation, Methodology, Visualization, Writing - original draft; Yi Wu, Conceptualization, Data curation, Methodology, Validation, Visualization, Writing - original draft; Yan Wang, Conceptualization, Data curation, Methodology, Visualization; Yubo Cui, Xiaosheng Huang, Sejie Yu, Conceptualization, Data curation, Investigation; Miao Zhang, Data curation, Investigation, Methodology; Tong Zhang, Data curation, Methodology, Visualization; Tao Yu, Conceptualization, Formal analysis, Methodology, Supervision, Writing – review and editing; Jun Zhao, Conceptualization, Formal analysis, Funding acquisition, Methodology, Project administration, Supervision, Validation, Writing – review and editing

## Author ORCIDs

Shaoyi Mei (iD) http://orcid.org/0000-0002-0684-3821
Yi Wu (iD) http://orcid.org/0000-0002-9907-9403
Tao Yu (iD) http://orcid.org/0000-0001-6017-4852
Jun Zhao (iD) http://orcid.org/0000-0002-7285-5812

## Ethics

Ethical approval was provided by the Medical Sciences Ethics Committee at Shenzhen Eye Hospital Affiliated to Jinan University, China (No.: 20161103005). The ethical protocol in this study was approved by the Independent Ethics Committee of Shenzhen Eye Hospital, in accordance with the tenets of the Declaration of Helsinki (1975). Written informed consent and consent to publish were received from all study participants prior to obtaining the human tissue samples. Samples were de-identified by removing the identifiable individual information such as name and identity number before researching.

## Decision letter and Author response

Decision letter https://doi.org/10.7554/eLife.71256.sa1
Author response https://doi.org/10.7554/eLife.71256.sa2

---

# Additional files

## Supplementary files

• Transparent reporting form

## Data availability

All data generated or analysed during this study are included in the manuscript and supporting files. Source data files have been provided for Table 2.

The following previously published datasets were used:

| Author(s) | Year | Dataset title | Dataset URL | Database and Identifier |
|---|---|---|---|---|
| Berry V, Ionides A, Pontikos N, Georgiou M, Yu J, Ocaka LA, Moore AT, Quinlan RA, Michaelides M | 2020 | The genetic landscape of crystallins in congenital cataract | https://gnomad.broadinstitute.org/gene/ENSG00000115020?dataset=gnomad_r2_1 | Genome Aggregation Database, r2_1 |

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
