## [Editor Report]

This manuscript will be of interest to readers in the field of eye development and pathology and ocular geneticists. Using a zebrafish model, the authors have identified a new autosomal dominant mutation in the kinase domain of the human phosphoinositide kinase gene PIKFYVE and assessed its impact on congenital cataract (clouding of the lens). They demonstrated that heterozygous or homozygous disruption of the PIKFYVE kinase domain, but not the overexpression of PIKFYVE G1943E, led to a cataract defect, suggesting the mechanism of haploinsufficiency rather than dominant-negative inhibition of PIKFYVE activity. The authors have presented strong evidence that mutations in the PIKFYVE kinase domain could lead to congenital cataract. Overall, the same pipeline of in vitro and in vivo genetic approaches could be applied to other human genetic disorders.

---

## [Decision Letter]

**Decision letter after peer review:**

Thank you for submitting your article "Mutations in PIKFYVE cause autosomal dominant congenital cataract" for consideration by *eLife*. Your article has been reviewed by 2 peer reviewers, and the evaluation has been overseen by a Reviewing Editor and Mone Zaidi as the Senior Editor. The reviewers have opted to remain anonymous.

Essential revisions:

There were a number of major strengths in the paper. Highlighted below are some essential points of revision in addition to the others listed by the reviewers.

1) A major issue is the replication of these genetic mutations in other cataract patients, even just another small family or a few more sporadic cases. There are many genetic cohorts available for cataract research. The replication would be necessary to establish the significance of the current research.

2) The rescue experiments should be further extended to characterize in detail the impact of PIKFYVE (p.G1943E) mutation (other PIKFYVE kinase domain deletions/mutations) in pikfyve-deficient zebrafish.

*Reviewer #1:*

Overall, the approaches and methodology used in the work is appropriate and the data are soundly interpreted. This manuscript describes a missense mutation (p.G1943E) in the human phosphoinositide kinase gene, PIKFYVE, in a family with autosomal dominant congenital cataract and examines its role in lens pathology using a zebrafish animal disease model – which is the strength of the work. However, another PIKFYVE mutation (although not in the kinase domain as described in the present work) has been previously linked to cataract in humans. This, along with the lack of assessment of the newly identified PIKFYVE (p.G1943E) mutation or other deletions/point mutations in the kinase domain's ability/inability to rescue lens defects in pikfyve-deficient zebrafish, reduces the overall novelty of these findings. Further, while Baf-A1 treatment is shown to rescue vacuole defects in pikfyve-deficient zebrafish here, certainly an encouraging finding from the therapeutic point of view, it should be noted that Baf-A1 had been previously described to rescue vacuole defects in COS-7 cells. Finally, the abstract/Discussion sections give the impression that Baf-A1 can be generally used in the treatment of congenital cataract – however, this conclusion cannot be drawn broadly because congenital cataract is a multifactorial disease.

Strengths:

The PIKFYVE mutation (p.G1943E) in the kinase domain is novel and it is described in a multi-generational family with autosomal dominant bilateral congenital cataract, with no detected impact on cornea (as previously described PIKFYVE mutations have been linked to corneal dystrophy).

The demonstration that pikfyve-deficient zebrafish exhibit lens defects is novel.

Characterization of vacuoles in pikfyve-deficient zebrafish lens to demonstrate that they are likely amphisomes (that arise from fusion of autophagosome and late endosome) is new.

Weaknesses:

Previously, the PIKFYVE mutation (p.P968Vfs23) in two individuals of a family has been described to develop "early" (albeit not congenital) cataract (PMCID: PMC3209427 – indeed, the authors acknowledge this in Discussion). However, this means that the association of a mutation in the PIKFYVE gene with cataract is not entirely novel. Thus, the novelty here principally lies in the nature of the mutation, i.e. impacting kinase domain, and the nature of the cataract, i.e. congenital cataract.

The rescue experiments should be further extended to characterize in detail the impact of PIKFYVE (p.G1943E) mutation (other PIKFYVE kinase domain deletions/mutations) in pikfyve-deficient zebrafish.

HLEB3 cells are lens epithelial in origin, and there is no discussion on their significance in modeling the defects that appear to be in the fiber cells in the lens. For example, it can be asked, how specific is the vacuole-defect with regards to the use of a particular cell line, i.e. will YM201636 induce this defect in multiple cell lines (as previously described, it does seem to have this effect in COS-7 cells) – and if so, what is the significance of these findings specifically in the context of lens cells.

While it is acknowledged that the following is challenging, identifying multiple families or multiple novel mutations in the PIKFYVE kinase domain (in human or other animal models) will serve to further strengthen the work.

Rescue assays involving the PIKFYVE (p.G1943E) mutation in pikfyve-deficient zebrafish would shed more light on the impact of the mutation. Or alternately, if the PIKFYVE (p.G1943E) mutation rescues the lens defects in pikfyve-deficient zebrafish to similar extents as the wild-type – how would this be interpreted. Similarly, rescue by other (non-kinase domain) mutations of PIKFYVE could be evaluated, this would help to strengthen the case that specific domains of PIKfyve might exert different functions and therefore are associated with different defects.

While Figure 3C, D suggest that pikfyve(delta8) heterozygote mutants exhibit higher number of vacuoles as compared to the wild-type at 5 dpf, it is not clear whether they also exhibit cataract as observed in the human cases.

There is no quantification of vacuoles in HLEB3 cells treated with YM201636 in Figure 3E.

Higher magnification images of Figure 4B, C would help evaluate the pikfyve-deficient lens defects.

Further screening in humans to increase the number/type of independent cases of PIKFYVE mutations (not necessarily PIKFYVE (p.G1943E), since that is established to be rare).

While it is acknowledged that the authors used zebrafish as a model, their claim of haploinsufficiency as the cause of pathology would be further strengthened by studies using multiple animal models. For example, a Pikfyve knockout mouse model have been generated and heterozygous animals have been shown to survive (PMCID: PMC3075686). For example, but not limited to, overexpression of PIKFYVE (p.G1943E) mutation in Pikfyve heterozygous mice could be informative in further discerning the haploinsufficiency vs. dominant negative impact of the mutation in a mammalian model.

*Reviewer #2:*

This manuscript aimed to determine and characterize the genetic causal factor of autosomal dominant congenital cataract in a large, multi-generational Chinese Korea pedigree. The authors performed whole exome sequencing followed by bioinformatics analysis in three affected and one unaffected individuals and identified a potential pathological non-synonymous variant in the PIKFYVE gene. This novel variant could affect the PIP kinase domain of the protein. The authors demonstrated that heterozygous or homozygous disruption of the PIKFYVE kinase domain, but not the overexpression of PIKFYVE G1943E, in zebrafish led to cataract defect in early stage, suggesting the mechanism of haploinsufficiency rather than dominant-negative inhibition of PIKFYVE activity. The zebrafish model with the loss of PIKFYVE function led to aberrant vacuolation – cataract related phenotype in lens cells. This aberrant vacuolation phenotype could be significantly alleviated by Baf-A1 by inhibiting V-ATPase. The authors have presented strong in vitro and in vivo evidence that mutations in the PIKFYVE kinase domain could lead to congenital cataract. Overall, the same pipeline of in vitro and in vivo genetic approach could be applied to other human genetic disorders.

Strength

This article comes with several strengths. First, the article is written clearly and easy to follow. Second, the experimental design and sample selection for whole exome sequencing is simple and clear. The bioinformatics pipeline is relatively clear, leading to the successful identification of the novel variant. Third, this study presented both in vitro cell culture and in vivo zebrafish-based functional validation to determine the pathological effect on cataract development. Fourth, the transgenic zebrafish model with the loss of kinase activity presented convincing cataract-related clinical phenotype data. Overall, this is a strong study to indicate the potential genetic cause of the identified potential pathological variant in the PIKFYVE gene.

Weakness

Despite the strength of this study, several weaknesses have been noticed as well. First, the authors need to establish the expression of PIKFYVE gene and protein in human lens capsules or lens cells. Second, the potential impact of the identified variant on the PIP kinase domain was mostly based on in silico prediction with protein structure. Additional data could be included to support the potential impact of this novel variant on the kinase activity in vitro. Third, the loss of PIP kinase activity via CRISPR needs to be confirmed using protein/mRNA expression data or kinase activity measurement in vitro. Fourth, the clinical phenotypes of transgenic zebrafish were only examined in early development and could be followed up for longer time past 7-dpf. Five, the identified rare variant has only been reported in one family. It will be more helpful to check whether other similar variants could be present in additional patients with congenital cataract.

1. Please spell the full name of PIKFYVE for the first time on page 3.

2. Please label the missing genotype information for all the individuals in the pedigree in Figure 1.

3. The identified variant should not be called "mutations" simply. The authors should follow the recommendations from ACMG to name the identified variant as "potential pathological variant".

4. It is necessary to validate the gene/protein expression of PIKFYVE in human lens tissue and cells if possible. This is necessary to provide additional support of its role in cataract.

5. Additional work/data is necessary to demonstrate the reduced or affected kinase activity related with the identified variant or the CRISPR-introduced kinase loss in zebrafish.

6, Please double check the manuscript for typos or grammar issues. For example, line 20 "affecting" should be replaced with "is located in".

---

## [Author Response]

Essential revisions:There were a number of major strengths in the paper. Highlighted below are some essential points of revision in addition to the others listed by the reviewers.1) A major issue is the replication of these genetic mutations in other cataract patients, even just another small family or a few more sporadic cases. There are many genetic cohorts available for cataract research. The replication would be necessary to establish the significance of the current research.

As suggested, we screened another congenital cataract family, 10 sporadic cases with congenital cataract and 200 cases of sporadic cataract patients for *PIKFYVE* variants by exon sequencing. Indeed, 6 affected loci in PIKFYVE from 7 patients were identified and two variants (p.Q1800P and p.A1865V) were found to be located in the PIP kinase domain of PIKFYVE (Figure 2—figure supplement 2 and Figure 2 supplement 2-source data 1). We have included these data in the result of revised manuscript (Line 129-135 on Page 7 and 8). These findings further support that PIKFYVE dysfunction is a causal/risk factor for cataract formation.

2) The rescue experiments should be further extended to characterize in detail the impact of PIKFYVE (p.G1943E) mutation (other PIKFYVE kinase domain deletions/mutations) in pikfyve-deficient zebrafish.

In the revised manuscript, we crossed the *Tg*(*ubi:PIKFYVE^G1943E^*) with *pikfyve^Δ8^* mutants and compared in parallel the rescue effects of wild-type and G1943E variant form of *PIKFYVE* in the mutants. Although *PIKFYVE^G1943E^* could also partially rescue the lens vacuolation defects in *pikfyve^Δ8^* mutants, the rescue effect of *PIKFYVE^G1943E^* was less efficient than that of *PIKFYVE^WT^* (Figure 3—figure supplement 3). These data have been included in the result of the revised manuscript (Line 172 to 180 on page 9 and 10). Meanwhile, as generation of stable transgenic lines are labor-intensive and time-consuming, to characterize the rescue effects of other PIKFYVE kinase domain deletions/mutations, we tried mRNA injection and utilized PIKFYVE^WT^ as the positive control. Unfortunately, transient injection of PIKFYVE^WT^ failed to rescue the lens vacuolation defects in *pikfyve^Δ8^* mutants (data not shown). Therefore, it is unclear to what extent other PIKFYVE kinase domain deletions/mutations could potentially rescue the *pikfyve*-deficient zebrafish. However, based on the rescue result of G1943E variant, we anticipate that other deleterious deletions/mutations could either not or less efficiently rescue the lens vacuolation defects in *pikfyve^Δ8^* mutants.

Reviewer #1:[…]Rescue assays involving the PIKFYVE (p.G1943E) mutation in pikfyve-deficient zebrafish would shed more light on the impact of the mutation. Or alternately, if the PIKFYVE (p.G1943E) mutation rescues the lens defects in pikfyve-deficient zebrafish to similar extents as the wild-type – how would this be interpreted. Similarly, rescue by other (non-kinase domain) mutations of PIKFYVE could be evaluated, this would help to strengthen the case that specific domains of PIKfyve might exert different functions and therefore are associated with different defects.

Thank reviewer 1# for the kind suggestions and our response to this issue can be found in the second part of *Essential Revisions.*

While Figure 3C, D suggest that pikfyve(delta8) heterozygote mutants exhibit higher number of vacuoles as compared to the wild-type at 5 dpf, it is not clear whether they also exhibit cataract as observed in the human cases.

In our zebrafish model, the *pikfyve* homozygous mutant fish show severe defects in lens development, and exhibit cataract phenotype. Meanwhile, the heterozygote fish also exhibit higher number of vacuoles as compared to the wild-type at 5 dpf. However, due to the small shape and limited number of these vacuoles as well as no obvious aggravation at later stages, it seems that these limited vacuoles in *pikfyve^Δ8^* heterozygotes has little effect on the transparency of the lens, which is possibly due to species difference between zebrafish and human. In fact, some human congenital cataract patients with lighter opacity also will not affect the transparency of the lens.

There is no quantification of vacuoles in HLEB3 cells treated with YM201636 in Figure 3E.

We have quantified the number of vacuoles in HLEB3 cells treated with YM201636 or DMSO. These results have been included in Figure 3F.

Higher magnification images of Figure 4B, C would help evaluate the pikfyve-deficient lens defects.

The magnified images have been included in Figure 4B and 4C.

Further screening in humans to increase the number/type of independent cases of PIKFYVE mutations (not necessarily PIKFYVE (p.G1943E), since that is established to be rare).

Thank reviewer 1# for the kind suggestion. Indeed we screened another 200 sporadic cases of cataract patients for other *PIKFYVE* variants and uncovered two genetic variants, which are located in the PIP kinase domain of PIKFYVE (Figure 2—figure supplement 2 and Figure 2 supplement 2-source data 1). We have included these data in the result of revised manuscript (Line 129-135 on Page 7 and 8). Our response to this issue can also be found in the first part of *Essential Revisions.*

While it is acknowledged that the authors used zebrafish as a model, their claim of haploinsufficiency as the cause of pathology would be further strengthened by studies using multiple animal models. For example, a Pikfyve knockout mouse model have been generated and heterozygous animals have been shown to survive (PMCID: PMC3075686). For example, but not limited to, overexpression of PIKFYVE (p.G1943E) mutation in Pikfyve heterozygous mice could be informative in further discerning the haploinsufficiency vs. dominant negative impact of the mutation in a mammalian model.

We highly agree with the reviewer’s opinion that the cause of pathology by p.G1943E variant could be further strengthened by studies with multiple animal models. We also thank the reviewer’s advice for the generation of mice model to investigate the pathogenesis of PIKfyve in cataract development. Although the Pikfyve knockout mouse has been reported in previous study and heterozygous animals have been shown to survive, due to the pandemic of COVID-19, it is difficult to import mouse strain from foreign countries to mainland China. On the other hand, generation of *Pikfyve*-knockout or -overexpressing mouse strains is also time-consuming and technical challenging for us. Therefore, we feel such a pity that we are unable to address the pathogenesis of PIKFYVE p.G1943E variant in mouse model at the moment.

Reviewer #2:[…]1. Please spell the full name of PIKFYVE for the first time on page 3.

We have added the full name in the revised manuscript (Line 56-57 on page 3).

2. Please label the missing genotype information for all the individuals in the pedigree in Figure 1.

We have labeled the genotype information in Figure 1.

3. The identified variant should not be called "mutations" simply. The authors should follow the recommendations from ACMG to name the identified variant as "potential pathological variant".

As suggested, we have changed the term “mutations” into “variants” accordingly in the revised manuscript.

4. It is necessary to validate the gene/protein expression of PIKFYVE in human lens tissue and cells if possible. This is necessary to provide additional support of its role in cataract.

We have collected the anterior lens capsules from three individuals with cataract during surgery and conducted quantitative RT-PCR to check *PIKFYVE* gene expression level. It showed that PIKFYVE was indeed expressed in human lens capsule (Figure 2—figure supplement 1). This data has been included in the result of the revised manuscript (Line 111-114 on page 6 and page 7)**.**

5. Additional work/data is necessary to demonstrate the reduced or affected kinase activity related with the identified variant or the CRISPR-introduced kinase loss in zebrafish.

We fully agree with the reviewer 2#’s opinion that measurement of the kinase activity would help clarify how the variants affect the protein function. Although the method has been developed recently, it is technical-challenging and time-consuming for us to set up this platform. Meanwhile, commercial detection kits are also unavailable. Therefore, we could only make assumptions of the change of kinase activity in PIKFYVE variants based on the *in-silico* prediction of protein structures at the moment.

6, Please double check the manuscript for typos or grammar issues. For example, line 20 "affecting" should be replaced with "is located in".

We really apologize for the typos or grammar mistakes in the manuscript. We have checked the manuscript thoroughly and revised those typos or grammar mistakes. In particular, we have changed the term “affecting” to “is located” (Line 21 on page 2).